# Physiological Effects of Microbial Biocontrol Agents in the Maize Phyllosphere

**DOI:** 10.3390/plants12244082

**Published:** 2023-12-06

**Authors:** María Fiamma Grossi Vanacore, Melina Sartori, Francisco Giordanino, Germán Barros, Andrea Nesci, Daiana García

**Affiliations:** 1PHD Student Laboratorio de Ecología Microbiana, Departamento de Microbiología e Inmunología, Facultad de Ciencias Exactas, Físico-Químicas y Naturales, Universidad Nacional de Río Cuarto, Ruta 36 km 601, Río Cuarto 5800, Córdoba, Argentina; mgrossi@exa.unrc.edu.ar; 2Consejo Nacional de Investigaciones Científicas y Técnicas (CONICET), Laboratorio de Ecología Microbiana, Departamento de Microbiología e Inmunología, Facultad de Ciencias Exactas, Físico-Químicas y Naturales, Universidad Nacional de Río Cuarto, Ruta 36 km 601, Río Cuarto 5800, Córdoba, Argentina; msartori@exa.unrc.edu.ar (M.S.); gbarros@exa.unrc.edu.ar (G.B.); anesci@exa.unrc.edu.ar (A.N.); 3Microbiology Student Laboratorio de Ecología Microbiana, Departamento de Microbiología e Inmunología, Facultad de Ciencias Exactas, Físico-Químicas y Naturales, Universidad Nacional de Río Cuarto, Ruta 36 km 601, Río Cuarto 5800, Córdoba, Argentina; giordaninof@gmail.com

**Keywords:** maize, biocontrol agents, phyllosphere, plant physiological response

## Abstract

In a world with constant population growth, and in the context of climate change, the need to supply the demand of safe crops has stimulated an interest in ecological products that can increase agricultural productivity. This implies the use of beneficial organisms and natural products to improve crop performance and control pests and diseases, replacing chemical compounds that can affect the environment and human health. Microbial biological control agents (MBCAs) interact with pathogens directly or by inducing a physiological state of resistance in the plant. This involves several mechanisms, like interference with phytohormone pathways and priming defensive compounds. In Argentina, one of the world’s main maize exporters, yield is restricted by several limitations, including foliar diseases such as common rust and northern corn leaf blight (NCLB). Here, we discuss the impact of pathogen infection on important food crops and MBCA interactions with the plant’s immune system, and its biochemical indicators such as phytohormones, reactive oxygen species, phenolic compounds and lytic enzymes, focused mainly on the maize–NCLB pathosystem. MBCA could be integrated into disease management as a mechanism to improve the plant’s inducible defences against foliar diseases. However, there is still much to elucidate regarding plant responses when exposed to hemibiotrophic pathogens.

## 1. Introduction

Maize (*Zea mays* L.) is one of the most important crops in the world and is used for human and animal consumption, as well as being a source of biofuel. This cereal is composed mainly of starch, but also supplies proteins and fatty acids and vitamins and minerals of great nutritional value [1]. Furthermore, it provides high levels of phenolic acids, flavonoids and carotenoids with antioxidant properties [2]. Argentina is one of the world’s largest exporters of maize. In 2021/2022, this country produced over 52 Mtn of maize [3].

One of the main factors contributing to reduced crop productivity is the occurrence of diseases, combined with an improper management [4]. Several maize diseases, particularly foliar ones, have a negative impact on photoassimilates production, which results in lower grain yield. In most cases, the extent of the disease depends on the environmental conditions, the pathogens involved, the host’s susceptibility and human intervention. In particular, the main factors that benefit fungal disease development are changes in sowing dates, reduced tillage, irrigation, intense and frequent precipitations during the summer months, poor monitoring and the presence of volunteer maize from previous harvests [5,6,7,8].

One of the most important foliar fungal diseases in maize is northern corn leaf blight (NCLB), caused by *Exserohilum turcicum*, a teleomorph of *Setosphaeria turcica* (Leonard and Suggs). In years with serious occurrence of NCLB, yield of the susceptible hybrids has been documented to decrease by about 40–50% [9]. In the field, mycelia and conidia of *E. turcicum* overwinter in crop residues and can be transported for long distances [10]. This pathogen is a hemibiotroph microorganism that spreads biotrophically at early stages of infection before shifting to a necrotrophic lifestyle [11,12]. Symptoms appear as grey elliptical lesions beginning on the lower leaves of the plant. As the disease progresses, susceptible plants become covered with the necrotic lesions that converge, giving it the appearance of being “burnt”. NCLB reduces maize yield by destroying the photosynthetically active area. Yield is also affected indirectly during the harvest because of stem breakage and rot, since the decrease in photosynthetically active area causes remobilization of carbohydrates from the stems to provide for the cob [10,13,14].

Cultural management of NCLB includes the application of fungicides, the selection of hybrids with genetic resistance, crop rotation and changes in the sowing date. The latter is often avoided in years when drought is forecasted, as an early date increases the risk of drought matching critical periods of the crop in the central area of Argentina [8]. The prevalent method for disease management in maize and other crops is the use of chemical fungicides. The fungicides used to control NCLB are mixtures of strobilurins and triazoles e.g., pyraclostrobin + epoxiconazole; azoxystrobin + cyproconazole; picoxystrobin + cyproconazole, among others [15]. A proper application should be performed at the initial stages of the disease in such a way that the critical stages of the crop are protected [8,16]. However, these chemicals are moderately hazardous, Class II and, to be effective, must constantly protect new plant leaves, increasing costs [17]. In addition, a chemical control practice may cause environmental problems [18] and health problems [19]. In addition, the economic damage thresholds for NCLB are only recent, and maize fungicide applications are generally decided on subjective criteria [9].

There is a need to generate new preventive strategies for the management of foliar diseases in crops of interest, like NCLB, in order to reduce fungicide application. This implies that we should adhere to an eco-friendly model of sanitary practices using natural substances typical of the ecosystem to be controlled. In this sense, new preventive strategies could study the capacity of native phyllosphere microorganisms with biofungicide potential. The phyllosphere can be considered an ephemeral habitat in which microorganisms are expected to multiply and use niches as the leaves expand [20]. These microorganisms interact in several ways, with each other and with the plant, by competition, mutualism, commensalism, antibiosis or plant hormone generation [21].

For this, biological control is an alternative strategy to the use of chemical compounds and involves the use of beneficial microorganisms for disease control. The term microbiological control agent (MBCA) applies to the use of antagonist organisms or natural products extracted from them to suppress disease [22]. There are several examples of MBCA application against phytopathogenic moulds [23,24,25,26,27]. In particular, refs. [26,27] searched for microorganisms that are antagonists of *E. turcicum* and can be obtained from the maize plant phyllosphere for the control of NCLB. These authors selected two of these antagonist isolates and applied them to maize plants during a field assay for blight control. Application of *Bacillus* spp. showed a reduction in the disease caused by *E. turcicum*, which was higher than 50% during 40 days with a significant increase in the grain yield compared to the untreated plants [26,27]. Therefore, the use of MBCA becomes a powerful management alternative aimed at minimising the yield losses caused by fungi, including those that cause foliar diseases, improving plant resistance to diseases. The mechanisms by which MBCA can be useful in disease control can be direct, by antagonism of the pathogen, by competition for nutrients or space, antibiosis, mycoparasitism or biofilm formation, or indirect, by inducing a state of resistance on the plant that enhances its defences through biochemical changes against further infections [28]. The latter represents a convenient strategy for the protection of new leaves. Unlike chemical pesticides with known modes of action, there are difficulties in understanding interactions involving MBCA, plants and pathogens.

To develop a successful foliar biofungicide, it is necessary to understand the mechanisms by which the biocontrol is executed in order to achieve effectiveness. In this sense, the leaf microbiome helps the plant against the attack of pathogens in an indirect manner by activating its defence mechanisms. In plants, there is an innate nonspecific immune system and an acquired or adaptive one, which are differentiated in specificity and memory of the response to the attacking agent [29]. The latter responds to changes or disturbances in the cellular structure caused by pathogens, symbiotic or free-living microorganisms, the application of exogenous chemical substances that act as elicitors, in fertilisation or against abiotic stress [30,31]. 

In the study of MBCAs controlling diseases, several biochemical indicators can be monitored to determine the physiological activities triggered in the host by the MBCA and/or pathogen that make disease development incompatible. For example, ref. [32] reviewed the modes of action of MBCA against the diseases in general, emphasising screening techniques, risk assessments and practical use. More recently, ref. [33] explored the progress made in the use of the biocontrol agents against fungal plant diseases. Regarding the phyllosphere habitat, in 2012, the authors of [34] compiled knowledge about microbial life in the phyllosphere. Other reviews, such as refs. [35,36] revised the investigations carried out into the epiphytic microbial communities. Figure 1 summarises the reviews cited and highlights the growing relevance of this area of research. However, reviews about MBCAs applied to control foliar diseases, particularly in maize, are scarce. Therefore, in this review we intend to summarise the possible changes in the physiological parameters of maize plants in response to the application of MBCAs against foliar diseases. These alterations can easily be measured through changes in biochemical compounds concentrations, such as phytohormones, secondary metabolites and reactive oxygen species, and offer a notion of the sanitary status of the plant during the interaction with the phyllosphere microbiome.

## 2. Plant Immune Response

A plant’s induced defences are stimulated once the pathogen enters the plant, and involve two staggered mechanisms: PTI (pattern-triggered immunity) and ETI (effector-triggered immunity) [37,39]. Defensive response may comprise hypersensitive responses and cell death, reactive oxygen species (ROS) generation, stomatal closure, cell wall reinforcement, production of secondary metabolites and pathogenesis-related (PR) proteins [40,41]. Natural selection may favour pathogens that avoid plant immunity, thus leading to a compatible interaction [39]. 

The different defence mechanisms in plants may be grouped as innate constitutive and basal resistance. In the event that a pathogen successfully avoids the constitutive defences and colonisation takes place, plants rely on inducible immune responses to avoid the disease progress [42]. This sort of defence requires pathogen recognition before deploying active response against the attacker [43]. Following the early signalling events activated by the pathogen attack, elicitor signals are often amplified through the generation of secondary signal molecules, such as salicylic acid (SA), ethylene (ET) and jasmonic acid (JA). In addition, the defence response in the plant–fungal interactions is also closely related to the accumulation of many secondary metabolites, such as flavonoids, phenolic compounds and phytoalexins [44,45]. Pathogen identification relies on the detection of conserved pattern-associated molecular patterns (PAMPs), particular broadly conserved molecules associated with a large range of pathogens, such as flagellin and chitin, by pattern-recognition receptors (PRR) set on the extracellular face of the host cell. This leads to PAMP-triggered immunity (PTI), a basal immune response effective against a broad spectrum of pathogens. PTI limits the pathogen growth by callose accumulation, cell wall strengthening, defence-related gene activation, ROS production, rapid calcium influx and phosphorylation cascades [35,38,44,46]. The activation of PTI also results in a growth inhibition, revealing the balance between growth and defence.

During coevolution, pathogens develop mechanisms to overcome plant defences and allow parasitism. In these cases, pathogens secrete effector molecules that inhibit or weaken PTI, enabling infection. At the same time plants have evolved the ability to recognise specific pathogen effectors using resistance (R) proteins that activate effector-triggered immunity (ETI), and normally result in ROS and calcium accumulation followed by hypersensitive response and cell death (Figure 2) [39,47].

ETI also triggers the biosynthesis of SA and the expression of PR proteins, activating systemic acquired resistance (SAR) and linking the basal to the inducible resistance [42,48]. SAR consists of priming events, mainly associated with large amounts of transcriptional reprogramming once the plant has been exposed to certain pathogens that lead to a much faster and stronger defence response both locally and systemically [49]. SAR induction involves the production of mobile signals that translocate to distant non-attacked tissues to prepare against further infections [50]. Non-pathogenic microbes can also mediate the plant defence response through induced systemic resistance (ISR) [51]. Both SAR and ISR constitute long-term systemic resistance against a broad spectrum of pathogens, but normally their actions are antagonistic, and their range of pathogens may differ. Their signalling pathways are often antagonistic, as SAR depends on the SA pathway, but ISR relies on ET and JA [27,42,52,53]. However, since both pathogens and MBCA are often detected by similar mechanisms in the host, the difference between SAR and ISR is not clear [33]

## 3. Phytohormones

Phytohormones are small molecules that act in a complex network to regulate plant growth and development, reproduction and response to the environment [54]. As plants lack specialised immune cells, they rely on hormones to integrate responses according to the environmental and developmental information [55]. To balance the inherent fitness cost of resistance against pathogens there is a fine-tuned crosstalk between phytohormones that aids the plant in adopting the appropriate defensive response to pathogens. On the other hand, pathogens have evolved to manipulate the immune signalling network to disrupt and avoid the plant defence response for their benefit [56,57]. According to [58], a plant’s association with MBCA improves plant health via several mechanisms, one of which is by participating in the phytohormones pathways. Figure 3 summarises the interaction between phytohormones during colonisation of pathogens and MBCA, and research carried out on maize foliar diseases.

### 3.1. Main Hormones Related to the Plant Infections: Salicylic Acid, Jasmonic Acid and Ethylene

SA is a phenolic compound that participates in plant biotic resistance, abiotic tolerance, thermogenesis, seed germination, flowering, senescence, stomatal closure, photosynthesis and many other processes [68,69]. There is a well-established positive correlation between resistance against biotrophic pathogens and endogenous levels of SA, though SA’s role in the defence against the necrotrophic pathogens is not fully understood [70]. However, hormone regulation in defensive mechanisms may differ among different plants. For instance, resistance against *Botrytis cinerea* in tomato is regulated by SA, but by JA and ET in tobacco [71]. In *Z. mays*, SA contributes to resistance against the *Colletotrichum graminicola* pathosystem [59]. Ref. [72] documented that SA levels were low in the uninfected maize plants but increase after the infection with *C. graminicola* and *Bipolaris maydis.* In addition, ref. [73] observed that the exogenous application of SA or its analogues triggered pathogenesis-related (PR) protein gene expression and thus resistance to several pathogens. Mutant plants with altered SA synthesis pathways are more susceptible to several pathogen infections. SA biosynthesis and expression of pathogen-related proteins are triggered by ETI, activating the systemic acquired resistance (SAR) as a consequence, thus linking basal to inducible resistance [42,48]. SAR consists of priming events, mainly associated with large transcriptional reprogramming, once the plant is exposed to certain pathogens that lead to a much faster and stronger defence response both locally and systemically [49]. SAR induction involves the production of mobile signals that translocate to distant non-attacked tissues to prepare against further infections [50]. The SA signalling pathway stimulates the expression of the pathogen-related proteins involved in the cell wall reinforcement, lysis of invading cells and a hypersensitive reaction leading to localised cell death [27,68,69], and is a key molecule involved in SAR elicitation. Although SA is not the mobile signal for SAR per se, it participates in the biosynthesis and induction of the signal molecule, and also induces the defence response [74]. SAR priming of the defence response leads to a heightened reaction to further infections, both at the infection site and in the distant non-attacked tissues [75]. The application of MBCA can improve plant health through SA production or biosynthesis stimulation in the plant. In this sense, ref. [76] found that endophytic bacteria associated with sunflower-produced SA in vitro and enhanced plant performance under stress. In addition, fungal growth in vitro was strongly inhibited in the presence of these strains.

JA is known for its role as a signalling molecule during necrotrophic pathogen attack, although there is evidence of JA inducing resistance to some biotrophic pathogens [55,77]. JA induces the expression of the defence-related genes, such as antioxidant and cell wall-degrading enzyme genes. It also contributes to the induced systemic resistance (ISR) mediated by beneficial microbes, which leads to a stronger or faster activation of cellular defences [53].

ET is a gaseous hormone that regulates multiple processes in the plants, from developmental to physiological functions. ET is produced at the infection sites, and aids ISR by activating defence reactions in nearby cells; due to its gaseous nature, ET is not limited by vascular tissues [51,78]. This hormone not only acts upon the presence of a pathogen but also is involved in the response to biotic and abiotic stresses [71]. On the other hand, it activates defence responses such as callose deposition [79], phytoalexin and ROS production [80]. Ref. [60] evaluated a genome-wide nested association mapping of 5000 inbred lines of maize for resistance to NCLB and identified two pathogen-related transcription factors of the ethylene response factor family associated with NCLB resistance in maize. These ethylene response factors are activated in response to necrotrophic pathogen attacks. Nevertheless, several pathogens are capable of producing ET to improve colonisation [71]. There is evidence that the hemibiotrophic and necrotrophic pathogens produce ET at the late stages of infection or deliver effectors that drive the plant to produce ET [51]. As a stress-induced hormone, ET significantly decreases plant growth and development. In this area of research, there are several studies of beneficial microorganisms involved in ET cleavage, thus sustaining plant growth [81].

Both ET, along with JA, are central pieces of ISR, based on priming for better defence rather than the direct activation of defence, thus enhancing the plant response with low fitness cost [61]. Non-pathogenic microbes can mediate the plant defence response through ISR [51]. Ref. [82] demonstrated that ISR in *Arabidopsis* spp. produced by *P. fluorescens* operates through the ET and JA pathways. Previously, ref. [83] showed that the capacity of *P. fluorescens* to trigger ISR in rice was ET/JA dependent.

The SA and JA/ET pathways are not individual but rather a crosstalk and, generally, their interaction is antagonistic. This has been proposed as a strategy to efficiently assign resources between growth and defence [43]. The SA-mediated defence is triggered following biotrophic pathogen infection whereas necrotrophic pathogens activate a different defence pathway regulated by JA and ET. As a result of antagonism between these pathways, a heightened biotroph resistance is often correlated with necrotroph susceptibility [84]. Some pathogens exploit this antagonism to overcome SAR. In addition, JA is one of the main elicitors of ISR, whereas SA intervenes in SAR [27,53]. Both SAR and ISR constitute a long-term systemic resistance against a broad spectrum of pathogens, but normally their action is antagonistic, and their range of pathogens may differ. Their signalling pathways are often antagonistic, as SAR depends on the SA pathway, but ISR relies on ET and JA [27,42,52,53]. However, since pathogens as well as MBCA are often detected by similar mechanisms of the host, the difference between SAR and ISR is not clear [33]. Ref. [62] proved that JA in maize is essential for immunity against soil-borne pathogens. In another study, ref. [63] found that *Pseudomonas putida* triggered ISR via JA rendered maize more resistant to *C. graminicola* anthracnose. This resistance was visible as reduced leaf necrosis and fungal growth in MBCA-inoculated plants.

SA, JA and ET not only interact among them regarding plant defence but also with other hormones, such as ABA, auxins and cytokinin. Auxins inhibit SA responses and promote JA signalling. On the contrary, cytokinins strengthen the SA response [85]. The interplay between ABA and SA is generally antagonistic, and ABA treatment usually leads to compromised resistance. Nevertheless, there is evidence that ABA can promote resistance according to the pathosystem in study [77]. This intricacy aids in fitting defence responses to maximum effectiveness against a broad spectrum of pathogens [75].

### 3.2. Auxins

Auxin levels can be altered by pathogens and induce changes not only in the plant organ structure, tumours and other growth anomalies but also other effects regarding colonisation that may not lead to altered growth or development. For example, indole-3-acetic acid (IAA) controls expansins, thus an increase in active IAA should render the cell wall more susceptible to penetration by a rising concentration of expansins, proteins that control cell wall loosening [86,87]. In addition, auxin and SA signalling have been proven to be antagonistic. Auxin suppresses SA-dependent defences [88,89], and on the contrary, SA-deficient plants show increased IAA levels [90]. Exogenous auxin can prevent PR protein production induced by SA [91]. On the other hand, auxin renders the plant more susceptible towards biotrophic and hemibiotrophic pathogens in both SA-dependent and SA-independent forms [87,88]. Moreover, JA and auxin signalling interact positively in most cases and share some similarities; as mentioned earlier, JA is antagonistic to SA [55,92]. Auxin-signalling plays an important role in defence against necrotrophic pathogens [93], which is consistent with JA’s role in necrotrophic pathogen-initiated response [94].

Pathogens can take advantage of the IAA pathway, either by synthesising auxin themselves or by inducing the plant’s auxin metabolism [95,96,97]. The amount of IAA synthesised due to the pathogen may influence tolerance in the host plant [98]. Thus, the excess of IAA will lead to the inactivation of the active auxin by negative feedback, therefore helping the plant in the resistance response while low levels of auxin might mimic host levels and not be detected by the host, increasing the disease symptoms [87]. Ref. [99] found differential expression of auxin-related genes in maize infected with the foliar pathogen *S. turcica*. In another study, ref. [100] discovered that seven auxin response transcription factors were strongly expressed in the resistant maize in response to infection with the foliar pathogen *Cercospora zeina*. Differential expression of auxin-related genes during the infection suggests that the auxin pathway may be involved in the defence response to fungal pathogens. Ref. [101] provided evidence that auxins inhibit PAMP-triggered ROS after corn smut by *Ustilago maydis* infection in maize. On the contrary, ref. [102] found that IAA inhibited growth in *Harpophora maydis*, a fungal soil-borne pathogen that causes late wilt in maize.

Auxin synthesis by beneficial bacteria can be one of many ways MBCA influence plant-pathogen interactions. Ref. [103] found that bacterial auxin reduced symptoms of head blight in barley by *Fusarium culmorum*; although, in vitro the hormone did not have any effect on pathogen growth. Endophytic *Bacillus* produces auxins among other phytohormones and increases nutrient intake by enhancing its accessibility to the plant [104]. Despite great progress over recent years, there is still much to be elucidated. However, there is strong evidence that auxin interacts with the SA and JA pathways, and an active auxin concentration leads to variations in susceptibility to different pathogens, depending on the pathogen lifestyle. Therefore, MBCA may affect the result of the plant-pathogen interaction by producing auxin or altering the plant’s auxin metabolism.

### 3.3. Abscisic Acid

Another relevant phytohormone is ABA, originally described as a growth-regulating and stress-response hormone, though there is evidence that it plays an important role as a modulator of the plant defence responses, most commonly as a negative regulator of the disease resistance [54,105]. Refs. [106,107] presented evidence of ABA’s negative role in the plant immunity. More recently, ref. [64] showed that the application of ABA on maize leaves produced enhanced *C. graminicola* disease progress. However, there are exceptions where ABA can positively regulate resistance in several pathosystems. Ref. [108] suggested that ABA participates as a chemical regulator of root-to-leaf SAR in maize, and proved that ABA treatment of roots reduced the fungal growth of *C. graminicola* on leaves, thus mimicking biological SAR. These results are consistent with [109], who provided evidence that ABA is involved in defence gene induction and resistance to the *S. turcica* pathogen. In another study, transgenic maize plants expressing *Lr34*, a gene involved in ABA transport that increases ABA levels in leaf tips of wheat and barley, showed enhanced resistance against common rust and NCLB [65]. Similar results were obtained by [110] when they applied exogenous ABA and reduced the spread of the fungus *Cochliobolus miyabeanus* in rice due to antagonistic crosstalk with ethylene.

In maize inoculated with *P. putida*, a rhizobacterium that colonises roots and has the ability to adhere to maize seeds, ref. [63] found that the bacterium elicited a response in maize that included ISR and was mediated by ABA and JA upregulation. When exposed to the pathogen *C. graminicola*, inoculated maize plants exhibited fewer disease symptoms and fungal growth in contrast to uninoculated ones. In summary, ABA can improve or impair plant defences depending on the plant–pathogen system and the timing of the infection rather than the pathogen lifestyle or plant species [111]. Early in the plant–pathogen interaction, increased ABA levels may stimulate host resistance, but can trigger the opposite effect once the pathogen has penetrated the host tissue, as ABA can interfere with ROS production [108]. The divergent effect of ABA on disease responses suggests an efficient defence regulation strategy by which ABA promotes physical barriers to early stages of colonisation and prevents the unnecessary activation of SA and JA-dependent defences [105].

In summary, plants need to survive and reproduce in a changing environment, involving biotic and abiotic factors that influence growth and development. In this sense, the cost of being well-defended impacts negatively on growth. Phytohormones aid in integrating environmental cues to achieve the best possible outcome. Three hormones, ET, SA and JA, are best known to mediate defence responses to biotic stresses such as pathogen attacks, and their effect is well known. However, there is evidence that other hormones are involved in varying degrees during pathogenesis, and that there are multiple signalling networks that help achieve the appropriate response for each scenario. In addition to the plant–pathogen interaction, MBCAs may influence hormonal homeostasis to render the plant more resistant to further infections. In the maize-*E. Turcicum* pathosystem we found evidence of SA, ET, auxins and ABA implication. 

### 3.4. Cytokinins

Increased cytokinin (CK) levels improve plant resistance to biotic and abiotic stresses, such as drought, salt and some diseases [112]. CK also promotes the production of antimicrobial compound phytoalexins [113] and interacts with SA [114]. Ref. [115] found that CK concentration positively correlates with the stress exposure and is a good indicator of resistance to different stresses. The influence of this phytohormone on immune response depends on interactions in a complex hormonal network. For example, CK modulates the SA pathway and promotes resistance to several pathogens. In addition, CK interacts synergistically with SA to activate PR gene expression, thus enhancing resistance to the disease [66]. The interaction between auxin and CK may result in augmented immunity or susceptibility as a result of a delicate balance. Ref. [85] reviewed the idea that increased vulnerability is reduced by CK, and solid resistance is diminished by auxin in the *Arabidopsis—P. syringae* pv. tomato pathosystem. CK’s role in defence could also be linked to defence-related gene priming [66,116].

However, it has been proven that CK activity, just like auxins, is exploited by some pathogens. Fungal CK play important roles in plant–pathogen interaction and disease development [117] and some pathogens also secrete CK or activate plant CK pathways to deviate nutrients from the host toward infected tissues [118]. The increased levels of CK produce “green islands’”, juvenile tissue by which certain pathogens create their own metabolic pool. Ref. [119] suggested that fungal modulation of CK metabolism affects host physiology. A hypothesis of this phenomenon is associated with a CK disorder caused by fungal infection [120]. Some works showed that final CK concentrations were much higher in infected susceptible hosts of barley and maize than the resistant variety for *Pyrenophora teres* and *Dreschslera maydis* infections, respectively [121]. Overall, CK influence over plant–pathogen interactions is complex and may differ according to the pathosystem, manifesting the result of coevolutionary interactions. Moreover, there is evidence that plant CK receptors may be able to recognize both plant and pathogen derived CK and elicit different outputs [66]. Several MBCAs can produce CK and improve plant growth and defence. Ref. [117] showed that *Trichoderma* strains can produce CK, and impact positively on *Arabidopsis* spp. resistance to *Fusarium graminearum*. Some bacteria, like *Azotobacter* spp., *Pantoea* spp., *Pseudomonas* spp., *Bacillus* spp. And *Paenibacillus* spp., produce CK [122]. So far, we have not found evidence of the role of CK in NCLB biocontrol. 

### 3.5. Giberellins

Gibberellins (GA) stimulate plant growth by promoting the degradation of DELLA proteins, a class of nuclear growth-suppressing proteins [67]. They were discovered when the rice disease fungus *Gibberella fujikuroi*, now *F. fujikuroi*, was isolated [123]. High amounts of fungal GAs produce abnormally elongated stems and suppression of plant GAs biosynthesis in rice [124]. These types of phytohormones are receiving more attention regarding their role in defensive response. Recent evidence suggests that GAs and DELLA proteins could be core participants in the defence response to pathogens, as there is a delicate crosstalk between GAs and other hormones that sustain the balance between growth and development. In this sense, GA can stimulate or suppress plant defence responses according to the plant-pathogen combination by crosstalking with SA and JA. Usually, GAs enhance resistance to biotrophs by activating the SA-dependent response and enhance susceptibility to necrotrophs by repressing the JA-dependent defence response [125,126]. ROS have a negative effect on disease resistance to necrotrophic pathogens [127], thereby an increase in GA might lead to the enhanced degradation of DELLA proteins, and this can lead to an augmented susceptibility to necrotrophic pathogens.

Ref. [124] studied the hormonal status in maize infected with several strains of *Fusarium* spp., and observed increased active GAs content after *Fusarium* infection, except for *F. verticilloides.* They also found GAs in axenic cultures of *F. fujikuroi* and *F. proliferatum*. MBCA may influence a plant’s hormonal balance by altering the GAs pathway to promote plant growth and defence. Ref. [128] proved that the plant-growth-promoting rhizobacteria *B. pumilus* and *B. licheniformis* produce GAs in vitro. These GAs are biologically active, as proved in *Alnus glutinosa* treated with bacterial media. In another study, ref. [129] presented a strain of *B. amyloliquefaciens* with the same ability, and its beneficial effects on rice plants. Ref. [130] found physiologically active GAs production in the endophytic *Sphingomonas* spp. pure culture. Tomato plants inoculated with this bacterium showed increased shoot length, shoot and root dry weight and chlorophyll content. These results confirm the abilities of plant growth-promoting bacteria through hormonal stimulation. Still, there is much left to discover regarding GAs role in plant disease biocontrol.

## 4. Secondary Metabolism Compounds: Phenolic Compounds and Phytoalexins

Phenolic compounds are secondary natural metabolites with significant diversity that can modulate crucial physiological processes such as transcriptional regulation, membrane permeability, vesicle trafficking and signal transduction, oxidative burst and photosynthesis rates [131]. During stress, plant tissues accumulate phenolic compounds that regulate ROS. Phenolic compounds also participate in the cell wall structure, as they are the monomers of lignin molecules. Furthermore, some phenolic compounds, like SA, participate in signal transduction pathways [132,133].

Phenols actively participate in plant defence through direct interference with pathogens and by the reinforcement of structural components to present a mechanical barrier [134]. Lignin acts as a physical barrier against fungal penetration, rendering the plant cell wall more resistant to mechanical penetration and restricting the diffusion of fungal toxins and enzymes [135]. Furthermore, phenolic acids contribute to plant defence against pathogens by ameliorating mycotoxin effects due to their antioxidant properties and in vitro ability to inhibit mycotoxin biosynthesis [136]. In maize, total phenolic content increases after pollination, as well as its antioxidant activity, both positively correlated traits [137]. Ref. [134] found that free chlorogenic and ferulic acid (two phenolics compounds) could be linked to maize defence against *F. graminearum* and that susceptibility may depend on biosynthesis in planta of these phenolic compounds. Ref. [138] found that the highest resistance to *F. verticillioides* in the maize cultivar which had the highest phenolic content. Regarding foliar diseases, ref. [99] associated a gene involved in lignin production and phenylpropanoid pathway to quantitative resistance to NCBL.

MBCA can prime host defences by enhancing phenolic production. In a study of *B. pumilus*, *F. oxysporum* spreading in pea roots was restricted when *B. pumilus* was present due to a physical barrier. This cell wall strengthening included callose and phenolic compounds beyond the site of infection with *B. pumilus*, suggesting systemically induced resistance [139]. Ref. [140] studied the effect of MBCA, along with the application plant extracts , in tomato response to *Alternaria solani* disease. The higher levels of phenolics, in addition to higher activity of defence related enzymes, such as antioxidant and lytic enzymes, resulted in induced resistance and higher yield. On the other hand, maize plants treated with *T. viride* and SA showed a higher activity of defence related phenols against NCLB [141]. In addition, ref. [142] found that *Bacillus* and *Pseudomonas* (MBCA) strains primed tomato plants for *Agrobacterium tumefaciens* infection by enhancing phenolic compounds synthesis of and increasing antioxidant enzymes activity. Further, in this study, SA content was higher in primed and infected plants with *A. tumefaciens*, suggesting the ability of some strains to trigger SAR.

Phytoalexins are chemically diverse, low antimicrobial weight compounds that are synthesised and accumulated in plants in response to biotic and abiotic stress. Phytoalexins inhibit pathogenic fungi and bacteria but are also toxic to other organisms. They can inhibit hyphal growth, sporulation and spore germination in fungi [143].

Maize phytoalexins were first isolated in 2011, and so far, three types have been identified: kauralexins, zealexins and benzoxazinoids [144,145]. These are strongly accumulated after infection by *C. graminicola*, *U. maydis*, *F. graminearum* and other pathogens, and after abiotic stresses such as drought. Such accumulation varies among inbred maize lines and hybrids, and is positively correlated with fungal disease resistance [146]. In liquid cultures, maize phytoalexins inhibit fungal growth at concentrations as low as 10 μg/mL [147]. 

Some MBCAs can stimulate phytoalexin production. We were not able to find evidence of phytoalexin-inducing MBCAs in maize, but there are studies that have been carried out in other crops. Phytoalexin production was higher in *Vigna* spp. inoculated with vesicular arbuscular mycorrhiza and, similarly, these plants had heightened tolerance to wilt disease [148]. More recently, ref. [149] found that one of the many ways in which *T. atroviride* provides resistance to southern corn leaf blight is by inducing phenylalanine ammonia lyase activity, a key enzyme in phenolics and phytoalexin synthesis.

## 5. Lytic Enzymes: Chitinase and β-1,3-Glucanase

β-1,3-glucanase and chitinases are types of pathogenesis-related proteins (PR) that hydrolyse fungal cell wall components. Their dual function is based on direct action by inhibiting fungal growth and indirectly by cleaving fungal walls to produce small subunits that act as elicitors for hypersensitive response [150]. These hydrolases act synergistically, both in vitro and in vivo [151]. In ref. [152], they identified seven chitinase genes associated with increased resistance to *Aspergillus flavus* infection and toxin accumulation in maize. A wide range of plant chitinases have been reported in sugarcane, rice, maize, wheat, tobacco, banana, sugarcane, *Arabidopsis* and *Sorghum bicolor* which are involved in defence mechanisms against fungal pathogens [58,153]. β-1,3-glucanase activation in response to pathogen infection has been studied in several crops. There is evidence that β-1,3-glucanase gene expression occurs early in compatible interactions but the quantity of the expressed gene decreases in incompatible interactions. Refs. [154,155] showed that *Fusarium* ear rot resistant maize lines present higher levels of expression of PR proteins such as glucanase and chitinase than susceptible ones. Ref. [156] found many more transcripts of β-1,3-glucanase and chitinase encoding genes in *Fusarium* ear rot-resistant maize seedlings infected with *F. verticillioides* than in susceptible ones. These results might indicate that a higher resistance could be associated with PR proteins. Moreover, wheat β-1,3-glucanase transcripts accumulated after treatment with SA, JA and ET, suggesting involvement in SAR or ISR [154].

Lytic enzymes from microorganisms can be used as MBCAs as part of an integrated pest management [157]. Ref. [158] found high activity of β-1,3-glucanase in *B. subtilis* isolates from the maize phyllosphere. This *Bacillus*, now identified as *B. velezensis* (Genbank OL704805), has been extensively studied demonstrating its MBCA capacity. Currently, a formulation is being developed to apply against NCLB disease [31,32,158]. Ref. [159] attributed the antagonist activity of *T. asperellum* against *F. graminearum* to cell wall degrading enzymes, among other factors. Ref. [160] isolated glucanase-producing rhizobacteria from wheat-maize cropping systems to decrease *Fusarium* wilt in tomatoes. A pH-, temperature- and salinity-stable chitinase is produced by a strain of *B. subtilis* induced by the pathogen *Botrytis cinerea* [161]. Additionally, MBCAs can prime the host plant to produce these enzymes. *Bacillus* spp. applied to primed soybean plants revealed higher defence-related enzymes, such as cell wall degrading and antioxidant enzymes, as well as higher JA and phenolic content [162].

## 6. Reactive Oxygen Species

Reactive oxygen species (ROSs) are unstable oxygen molecules normally produced in plant cells as a result of normal oxygenic metabolism [58]. At regular levels, ROSs participate in physiological processes such as programmed cell death and senescence, but higher levels of ROSs can cause toxicity by disrupting membranes, proteins and nucleic acids, and this ability is exploited in resistance against pathogens [163]. During biotic stress due to disease, plants trigger a second level of defence after pathogen penetration through physical barriers. This step involves phenolic compounds synthesis, callose deposition and ROSs generation. ROSs are capable of killing the pathogen directly or inducing antimicrobial compounds biosynthesis [60], but they also can act as signalling molecules [58]. The hypersensitive response (HR) includes oxidative bursts involving high amounts of ROS and expression of PR genes [105]. Cell death associated with disease leads to chlorosis and dehydration, restricting the pathogen to the entry area and preventing proliferation [164]. This hypersensitive response is induced by incompatible pathogens and is a form of programmed cell death that requires active transcription [165]. However, some pathogens have developed mechanisms to overcome ROSs-associated immune response. For example, *U. maydis* effectors interact with maize peroxidases to scavenge ROSs, leading to biotrophic interaction establishment [166]. 

It is necessary to maintain ROSs levels under control to prevent their toxicity. Plants have both enzymatic and non-enzymatic mechanisms to counteract deleterious effects of ROS. The first include superoxide dismutase, ascorbate peroxidase and catalase, and the latter consist of antioxidant compounds like carotenoids, tocopherols and ascorbic acid [163,167]. Ref. [99] found that the expression levels of genes involved in ROSs scavenging changed in maize exposed to *S. turcica*, and measured higher antioxidant enzyme activity in resistant maize seedlings than in susceptible ones. These results might suggest that maintaining an ROSs balance could be related to maize resistance against NCLB. In addition, MBCA may aid in defence response by interfering in ROSs metabolism. Ref. [168] proved that *Trichoderma* spp. induced higher ROSs accumulation in maize exposed to downy mildew caused by *Peronosclerospora* spp. and reduced disease intensity (Table 1).

## 7. Research Gaps and Commercialization

In spite of the thorough investigation carried out on MCBAs applied to maize [169,170,171,172,173], the number of registered MCBA products is low and most of them involve the use of *Trichoderma* spp. and *Bacillus* spp., like *T. longibrachiatum* and *T. asperelloides* used for vascular wilt disease control in maize, which is caused by *Magnaporthiopsis maydis*, or the multiple applications of the genus *Bacillus* spp. in agriculture. One of the reasons for this is that in vitro studies do not correlate with in vivo field trial results. In this sense, causes may be due to production, formulation and delivery conditions as well as survival of MBCA in the field [174,175].

Another factor involved is commercialization. In this sense, the needs and practices of farmers must be taken into account to meet target consumer preferences. In addition, the registration process for new products is time and resource consuming. For this reason, several global agencies such as the Organization for Economic and Co-Operative development, EPPO and International Organization for Biological Control were created to overcome the regulatory hurdle and enhance commercialization [174]. The development of a successful MBCAs includes the involvement not only of plant pathologists, agronomists and microbiologists, but also of statisticians and marketers [176].

However, according to [172], global biopesticides sales increase 10% annually, and a reason for this is that environmental concerns and restriction of chemical pesticides generates a demand for safer and environmentally friendly products [174]. Ref. [176] explored the acquisition and joint ventures of biological companies by larger agrochemical companies, indicating a great investment interest into the biological industry.

Latin America accounts for only 10% of the global biopesticides market [177], despite the importance of agricultural products in its economy. For this reason, there is a great potential for the development of MBCA products and their extensive use in an integrated pest management program.

## 8. Conclusions

The current productive model involves the use of practices detrimental to the environment, and threatening to human health, in the context of increasing demand and climate change. Thus, biological control represents an alternative to modern pesticides. As can be seen throughout this review, there are several lines of research regarding the use of MBCAs to prevent or eliminate pathogens in different crops. However, to the best of our knowledge, scarce research has been carried out on the physiological response of the plants after application of these biocontrol organisms. Our particular interest is focused on physiological responses of maize plants due to the application of the potential MBCAs isolated in our lab to control *E. turcicum*, a causal agent of NCBL. In this review, we summarise some of the mechanisms by which plants defend themselves against pathogens and by which biocontrol is executed, although there is much to elucidate. The barriers that pathogens encounter during colonisation in plants are structural and inducible defences. Among the latter, signalling molecules play a major role by maintaining a fine-tuned network with differential responses depending on the pathogen involved. These molecules can also be redistributed to the plant and prime tissues that are distant from the infection. MBCAs can be implemented as a tool in integrated pest management in agroecosystems by taking advantage of inducible defences. Whilst we reviewed evidence of various interactions between MBCAs and plants in resistance against disease establishment, there is still much to unravel with respect to the plant physiological response. Studies implemented in real ecosystems should be carried out. For this, we will continue our investigations and aim towards the detection of biochemical compounds that are activated in maize plants by the application of potential MBCAs that are antagonistic to foliar diseases. 

## Figures and Tables

**Figure 1 plants-12-04082-f001:**
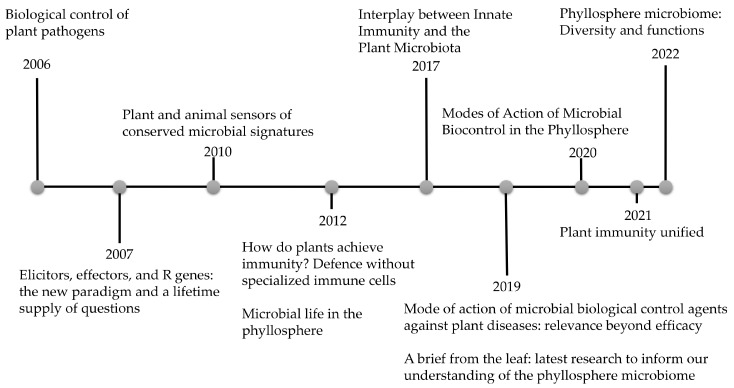
Research reviews on biological control, plant immunity and phyllosphere microbial communities through the years [22,28,29,30,31,32,34,35,36,37,38].

**Figure 2 plants-12-04082-f002:**
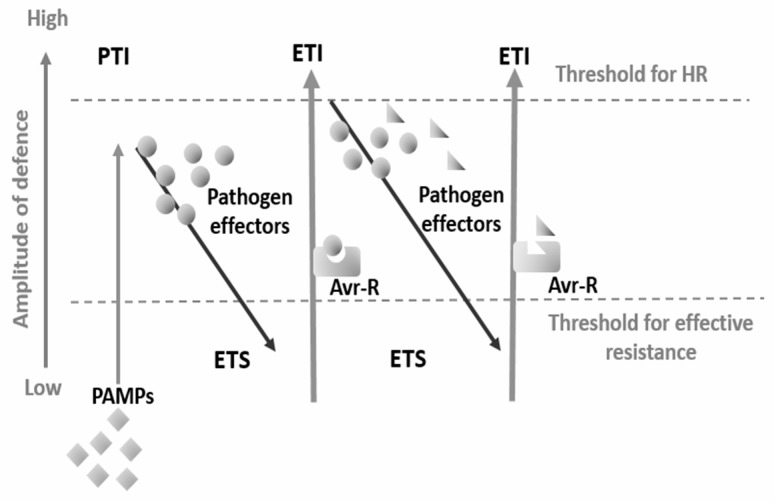
Plant immune system “zig-zag” model proposed by Jones and Dangl [39]. In the first phase, PAMPs are recognised by pattern recognition receptors, triggering PTI. Pathogens secrete effectors (circles) to avoid plant defence mechanisms and deploy ETS. Resistance proteins (Avr-R) that can recognise effectors activate ETI, a phase that usually trespasses the threshold for hypersensitive reaction (HR) and cell death. In the last phase, pathogens may develop new effectors (triangles) to suppress ETI. Natural selection favours new R protein alleles able to recognise these acquired effectors.

**Figure 3 plants-12-04082-f003:**
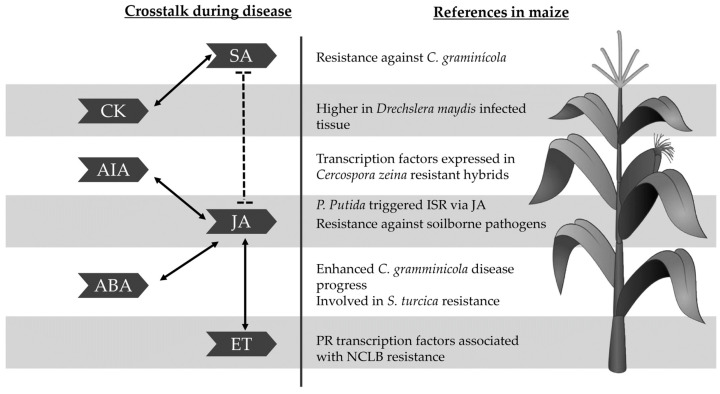
Crosstalk of phytohormones during pathogen and MBCA interaction, and research caried out on maize [43,55,59,60,61,62,63,64,65,66,67].

**Table 1 plants-12-04082-t001:** Summary of host plants, MBCAs and their mode of action against corresponding pathogens.

Host Plant	MBCA	Mode of Action	Pathogen/Disease	Reference
*Arabidopsis* spp.	*Pseudomonas fluorescens*	ISR via ET and JA	-	[82]
Rice	*Pseudomonas fluorescens*	ISR via ET and JA	*Magnaporthe oryzae*	[83]
Maize	*Pseudomonas putida*	ISR via JA and ABA upregulation	*Colletotrichum graminicola*	[63]
Pea	*Bacillus pumilus*	Phenolic compounds	*Fusarium oxysporum*	[139]
Maize	*Trichoderma atroviride* and SA	Phenolic compounds	*Exserohilum turcicum*	[141]
Tomato	*Bacillus* spp. and *Pseudomonas* spp.	Phenolic compounds and antioxidant enzymes	*Agrobacterium tumefaciens*	[142]
Maize	*Trichoderma atroviride*	Phenolic compounds and phytoalexins	Southern corn leaf blight	[143]
*Vigna* spp.	Vesicular-arbuscular mycorrhiza	Phytoalexins	Wilt	[148]
Soybean	*Bacillus* spp.	Cell-wall degrading and antioxidant enzymes and phenolic compounds		[162]
Maize	*Trichoderma* spp.	ROS accumulation	*Peronosclerospora* spp.	[168]

## Data Availability

No new data were created or analyzed in this study. Data sharing is not applicable to this article.

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
