# Peer review of "Physiological Effects of Microbial Biocontrol Agents in the Maize Phyllosphere"

_plants, 2023, doi:10.3390/plants12244082_

Round 1
Reviewer 1 Report
Comments and Suggestions for Authors
Dear Authors,
The review needs extensive revision for the English language. It seems to be translated by Google translate or another software. I have tried to go through the review but it is difficult to do in this case. The reference need to be updated. In addition, you have to add a part on the molecular aspects of the triggered plant defensive responses. In this form, the manuscript is unsuitable for publication in Plants.
Comments on the Quality of English Language
The review needs extensive revision for the English language. It seems to be translated by Google translate or another software. I have tried to go through the review but it is difficult to do in this case. In this form, the manuscript is unsuitable for publication in Plants.
Reviewer 2 Report
Comments and Suggestions for Authors
The review “Physiological effects of biocontrol agents in the maize phyllosphere” compile an important topic and can be published after following corrections.
1. In title replace biocontrol agent with microbial biocontrol agents and also throughout the MS.
2. Abstracts looks very primitive add findings and conclusion.
3. Keywords should be repeat title.
4. In introduction, write earlier published reviews on this topic and how your review is novel in this regard, also add a graphical image summarising whole story of this paper.
5. Line 115 avoid words like “try” moreover English correction and choose of the words are required for this review.
6. Clear objectives of this review are missing.
7. Line 174 to 190 reduce general statements/redundancy that nothing to do with this review. As I can see most of the sentences are general.
8. Provide a graphical image summarising the content under phytohormone section.
9. Write a separate section on research gaps and how these microbial agents can be commercialized and recommended to maize growers.
Comments on the Quality of English LanguageThe review “Physiological effects of biocontrol agents in the maize phyllosphere” compile an important topic and can be published after following corrections.
1. In title replace biocontrol agent with microbial biocontrol agents and also throughout the MS.
2. Abstracts looks very primitive add findings and conclusion.
3. Keywords should be repeat title.
4. In introduction, write earlier published reviews on this topic and how your review is novel in this regard, also add a graphical image summarising whole story of this paper.
5. Line 115 avoid words like “try” moreover English correction and choose of the words are required for this review.
6. Clear objectives of this review are missing.
7. Line 174 to 190 reduce general statements/redundancy that nothing to do with this review. As I can see most of the sentences are general.
8. Provide a graphical image summarising the content under phytohormone section.
9. Write a separate section on research gaps and how these microbial agents can be commercialized and recommended to maize growers.
Reviewer 3 Report
Comments and Suggestions for Authors
Dear Authors,
Paper contains many interesting information about physiological plants responses to BCA application.
It is a shame that none of the biological tests were taken to prove/confirm some of these considerations.
It could be good to expand biological control agents part of the paper by adding some more examples of BCA registred/tested (by other authors) in the corn production. And add some information about the benefits that comes from its application. Even these which are hipotethically dedicated to corn production according to its wide spectrum of controlled pathogens.
Maybe it could be good to change the title to fit more to the described considerations?
You can also add some text about the need of BCA application according to many withdrawals of active ingredients (Mainly in EUROPE) not only in corn cultivation. This should improve the paper and add some clarification why do we really need to take care of BCA in many crops and what are the benefits of using it vs chemical compounds.
Round 2
Reviewer 1 Report
Comments and Suggestions for Authors
Dear authors,
Please find attached the track change file.

Author Response
Dear reviewer 1,
Thank you very much for taking the time to review this manuscript. Please find the detailed responses below and the corresponding corrections highlighted in the re-submitted file.
- Line: 47 Change “benefit” to “lower”.
Here we are talking about the factors that increase the disease extent and benefit fungal development, such as intense and frequent precipitations. We considered changing the term to “benefit fungal diseases development” for a better understanding.
- Citation names in the text.
Journal instructive for manuscript presentation included the type of citation we used in our manuscript.
- Line 269. Example of SA and JA crosstalk in banana.
The interaction depends on the stress but in our manuscript we focus in SA and JA interaction during foliar fungal diseases, not root mycorrhizal colonization. In these cases, almost all of the references found indicate an antagonistic interaction between these phytohormones.

Reviewer 2 Report
Comments and Suggestions for Authors
authors addressed my all comments the MS can be accepted.
Comments on the Quality of English Languageauthors addressed my all comments the MS can be accepted.
Author Response
Dear reviewer 2,
Thank you very much for taking the time to review this manuscript. Some corrections of English language were made. We did not find another attached file with other corrections to the manuscript.
